# Patient-Led, Technology-Assisted Malnutrition Risk Screening in Hospital: A Feasibility Study

**DOI:** 10.3390/nu16081139

**Published:** 2024-04-12

**Authors:** Shelley Roberts, Andrea P. Marshall, Leisa Bromiley, Zane Hopper, Joshua Byrnes, Lauren Ball, Peter F. Collins, Jaimon Kelly

**Affiliations:** 1School of Health Sciences and Social Work, Griffith University, Southport, QLD 4222, Australia; zane.hopper@health.qld.gov.au; 2Allied Health Research, Gold Coast Hospital and Health Service, Southport, QLD 4215, Australia; 3School of Nursing and Midwifery, Griffith University, Southport, QLD 4222, Australia; a.marshall@griffith.edu.au; 4Nursing and Midwifery Education and Research Unit, Gold Coast Hospital and Health Service, Southport, QLD 4215, Australia; 5Nutrition and Food Services, Gold Coast Hospital and Health Service, Southport, QLD 4215, Australia; leisa.bromiley@health.qld.gov.au; 6Centre for Applied Health Economics, Menzies Health Institute Queensland, Griffith University, Southport, QLD 4222, Australia; j.byrnes@griffith.edu.au; 7School of Medicine and Dentistry, Griffith University, Southport, QLD 4222, Australia; 8Centre for Community Health and Wellbeing, The University of Queensland, St Lucia, QLD 4072, Australia; lauren.ball@uq.edu.au; 9Faculty of Medicine and Health, Sydney Nursing School/Susan Wakil School of Nursing and Midwifery, The University of Sydney, Sydney, NSW 2006, Australia; peter.collins@sydney.edu.au; 10Charles Perkins Centre, The University of Sydney, Sydney, NSW 2006, Australia; 11Centre for Online Health, The University of Queensland, Woolloongabba, QLD 4102, Australia; jaimon.kelly@uq.edu.au; 12Centre for Health Services Research, The University of Queensland, Woolloongabba, QLD 4102, Australia

**Keywords:** malnutrition risk screening, electronic nutrition screening, patient self-screening, patient-led screening, hospital malnutrition

## Abstract

Malnutrition risk screening is crucial to identify at-risk patients in hospitals; however, screening rates can be suboptimal. This study evaluated the feasibility, acceptability, and potential cost-effectiveness of patient-led, technology-assisted malnutrition risk screening. A prospective multi-methods study was conducted in a 750-bed public hospital in Australia. Patients were recruited from seven wards and asked to complete an electronic version of the Malnutrition Screening Tool (e-MST) on bedside computer screens. Data were collected on feasibility, acceptability, and cost. Feasibility data were compared to pre-determined criteria on recruitment (≥50% recruitment rate) and e-MST completion (≥75% completion rate). Quantitative acceptability (survey) data were analyzed descriptively. Patient interview data were analyzed thematically. The economic evaluation was from the perspective of the health service using a decision tree analytic model. Both feasibility criteria were met; the recruitment rate was 78% and all 121 participants (52% male, median age 59 [IQR 48-69] years) completed the e-MST. Patient acceptability was high. Patient-led e-MST was modeled to save $3.23 AUD per patient and yield 6.5 more true malnutrition cases (per 121 patients) with an incremental cost saving per additional malnutrition case of 0.50 AUD. Patient-led, technology-assisted malnutrition risk screening was found to be feasible, acceptable to patients, and cost-effective (higher malnutrition yield and less costly) compared to current practice at this hospital.

## 1. Introduction

Malnutrition is highly prevalent in hospitals, affecting 30–50% of inpatients [1]. Malnutrition is associated with poor patient outcomes, including increased risks of morbidity and mortality [2] and poor hospital outcomes, including increased length of stay, increased readmissions and greater costs [3,4]. The early detection of patients who are malnourished or at risk is crucial, not only to provide appropriate nutrition care for their prevention and treatment but to adequately recognize care costs in case-mix funding [5] and hospital-acquired complication reporting [6]. Research shows that early detection and intervention for malnutrition can reduce hospital costs and length of stay [7,8] and improve clinical outcomes, including survival, among malnourished patients [9]. Conducting malnutrition risk screening during hospitalization is an effective strategy to initiate early nutritional interventions and is recommended in international clinical practice guidelines [10].

Malnutrition risk screening, also referred to as nutrition screening, is the initial precursor step in the Nutrition Care Process [11]. It involves using a validated tool to identify patients who have or are at risk of developing malnutrition and who would, therefore, benefit from targeted nutrition care. Economic models show that identifying and managing malnutrition in hospitals is highly cost-effective; in England, the estimated costs of screening, assessment and nutritional support were more than offset by decreased healthcare use and costs, with a net saving of approximately GBP 65 million annually [12]. Clinical practice guidelines, therefore, recommend screening all patients for malnutrition risk upon hospital admission and weekly thereafter [10]. In many countries including Australia, hospitals mandate nutrition screening according to these guidelines [13]; yet, suboptimal screening rates are widely reported and malnutrition remains grossly under-recognized [14]. A study of 370 wards across 56 Australian and New Zealand hospitals found nutrition screening was routinely performed in just 64% of wards and routine re-screening in 14% of wards [15]. A study of 53 European hospitals found <40% of participating departments reported using a validated screening tool as part of routine practice and those that did had significantly lower prevalence of malnutrition, higher rates of dietitian referral, and more nutrition interventions implemented for patients than departments that did not routinely screen patients for malnutrition risk [16]. Clearly, further attention to malnutrition screening procedures is needed.

In Australia, nurses typically conduct nutrition screening. However, they face many barriers to completing this task, such as competing priorities and high amounts of administrative burden [17]. Low agreement occurs between dietitian- and nurse-completed screening [18] and staff report low levels of trust in accurate and timely screening by nurses [19]. A potential solution to these issues is involving patients themselves in the nutrition screening process. The Malnutrition Screening Tool (MST) is a quick, simple and reliable tool consisting of two yes/no questions and has been validated for use by medical, nursing, allied health and administrative staff, as well as by family/friends and patients themselves [20]. In fact, several studies have demonstrated the validity and reliability of patient-led nutrition screening [21,22], including electronic versions of screening tools [23], presenting a novel opportunity for patients (and/or families) to participate in nutrition care, an area in which patients indicate they feel comfortable participating [24].

Technology is revolutionizing healthcare and presents a myriad of opportunities to engage patients and streamline care. Several studies have demonstrated the feasibility, acceptability, and effects of using digital solutions for identifying malnutrition and managing nutrition care in hospitals [25,26], including by engaging the patients themselves in the nutrition care process [27]. Involving patients in their own nutrition risk screening via technology has many potential benefits. When patients participate in care, they have better clinical and functional outcomes [28], receive safer care they are more satisfied with, and experience higher self-efficacy [29]. Allowing patients to complete simple tasks releases staff time for other clinical activities, which is a major benefit in busy hospital environments. High agreement (≥94%) exists between electronic patient-led nutrition screening and paper-based health professional-led screening, with electronic self-screening being 40% faster [23]. There may be other benefits, such as improved referral processes, with flow-on effects for patients and hospitals if malnutrition is addressed earlier or more efficiently. Based on these preliminary findings and associated hypotheses, further research in this area is warranted.

The aim of this study was to evaluate the feasibility, acceptability, and potential cost-effectiveness of patient-led, technology-assisted nutrition screening in hospitals.

## 2. Materials and Methods

### 2.1. Study Design and Outcome Measures

This study used a prospective multi-methods design to evaluate patient-led technology-assisted nutrition screening using an electronic version of the MST (e-MST). The e-MST was accessed through the patient’s bedside computer screens and contained the same questions and wording as the validated MST tool [20]. Feasibility was assessed against the following criteria set a priori:(a)≥50% of eligible patients (or family member/s) approached agree to participate(b)≥75% of participants complete e-MST within 24 h of recruitment

Acceptability was assessed quantitatively through patient satisfaction surveys and qualitatively through interviews with a subset of participants.

Secondary outcomes relate to the economic and clinical implications of patient-led screening. The cost-effectiveness of this approach was evaluated in an economic sub-study estimating the incremental cost per additional true case of malnutrition identified with e-MST self-screening compared to usual care. The number and proportion of patients scoring as ‘at risk’ on the MST, the accuracy of nurse-completed vs. patient-completed MSTs, and the personnel time taken to engage patients in completing the e-MST are also reported. Finally, patient uptake was assessed in a separate phase of feasibility testing with minimal researcher prompting to complete the e-MST.

### 2.2. Study Setting and Participants

This study was conducted at a large (750 bed) tertiary public teaching hospital in Queensland, Australia. At this hospital, usual care involved ward nurses completing malnutrition risk screening within eight hours of admission for all patients, as per hospital procedure, using an MST form embedded in the hospital’s electronic medical records system. Patients with an MST score of two or more are automatically referred to the ward dietitian via this system. Across the wards, dietitians have limited access to dietetic assistants, who assist with some nutrition care-related tasks. The hospital had an electronic food service system (Delegate Technology GmbH, Vienna, Austria) in place, whereby patients accessed their menus and ordered their meals via bedside computer screens.

Patients were recruited from seven medical and surgical wards at the study hospital. Patients were eligible for recruitment if they met all inclusion criteria (aged ≥18 years, able to communicate in English, cognitively intact) and no exclusion criteria (prior participation in the study, dying or palliative, or diagnosed with an eating disorder). Family members, broadly defined as a relative, friend or carer of the patient, could participate on the patient’s behalf if they met the family inclusion criteria (aged ≥18, able to communicate in English, cognitively intact). During the consent process, a trained research assistant (RA) described the e-MST to patients and showed them how to access it via their bedside computer screen. Patients were informed that they could request assistance from the RA to complete the e-MST (these data were recorded). While there is no set sample size for feasibility studies [30], we expected to recruit around 100 patients and/or family members across the six wards, to provide adequate data for feasibility, acceptability and cost effectiveness assessments.

### 2.3. Data Collection

Table 1 outlines the data collection schedule. Participant demographic and clinical information were extracted from electronic medical records including age, sex, diagnosis, comorbidities, admission ward/date, MST score (completed by nurses) and dietetic input (including prescribed diet and supplements, and Subjective Global Assessment score if available).

Feasibility data were collected by the RA via a recruitment log (number of patients approached and consented) and digital e-MST reports (e-MST completion and timing) were extracted from the hospital’s electronic food service system. The time spent explaining/assisting patients with the e-MST was also recorded by the RA. Acceptability data were collected using a brief patient satisfaction survey (all participants; Appendix A) and interviews using a semi-structured interview guide (a subset of participants; Appendix A).

Cost data were collected from the electronic medical record for MST score, Subjective Global Assessment (SGA) score and malnutrition diagnosis; the electronic foodservice system for the e-MST score and clinician surveys of time spent on nutrition screening. Model inputs included e-MST diagnostic yield (additional patients identified as at-risk (MST score ≥ 2)), nurse time spent completing MST, RA time showing patients how to complete e-MST, dietetic time spent rescreening suspected false negatives (i.e., if nurse MST ≤ 1), unnecessary MST referrals (false positives), and malnutrition risk (yes/no). To determine MST false positives, dietitians on study wards were surveyed about which referrals were appropriate (yes/no).

To gain insight into patient uptake with minimal researcher prompting, a second phase of data collection involved sending an A5 flyer on patient meal trays with instructions on how to access and complete the e-MST. This was conducted on one ward per day, to see how many patients would complete the e-MST without being formally recruited to the study and receiving a direct prompt from a researcher to complete. Data were collected on the number of patients completing the e-MST and how many sought assistance, which was available from the RA upon request. This component was ethically approved as not requiring written consent from patients.

### 2.4. Data Analysis

All quantitative data were entered into IBM SPSS Statistics for Windows (IBM Corp: Armonk, NY, USA) for analysis. Continuous data were tested for normality using the Shapiro–Wilk test and presented as mean ± SD (normally distributed) or median and interquartile range (IQR; non-normally distributed). Feasibility data were presented as frequency and percent for comparison against feasibility criteria. Patient satisfaction survey data were also presented using frequency and percent. Agreement between patient- and nurse-completed MSTs was assessed using a Kappa test, interpreted with Cohen’s suggested ranges (values of ≤0 indicating no agreement, 0.01–0.20 none to slight, 0.21–0.40 fair, 0.41–0.60 moderate, 0.61–0.80 substantial, and 0.81–1.00 almost perfect agreement) [31].

Patient interviews were transcribed and analyzed using inductive thematic analysis per Braun and Clarke [32], including data familiarization, code generation, identification and review of themes, naming and describing themes, and writing up of findings with supporting quotes.

A health economist (JB) led the economic evaluation with assistance from a senior research fellow (JK), performed from the perspective of the health service using a decision analytic model (decision tree). For the economic analysis, it is important to understand our definitions of standard practice and the observed current practice in relation to MST screening and dietetic referral. Standard practice is defined as nurse-led MST screening as per the hospital’s screening procedure. If a patient scores MST 2 or more, they are automatically referred to dietetics. In the economic analysis for standard practice, we did not include the dietitian time spent screening their ward lists to identify patients who would not be detected via MST (i.e., those not yet considered at-risk) but who dietitians identify as likely needing nutrition support during their admission based on their diagnosis, as it is standard practice for dietitians to do this. As this is unrelated to MST screening, it was not accounted for in this analysis. Observed current practice is the practice we observed in this study, and reflects additional screening performed by dietetics due to low trust in nurses’ MST scores at the hospital. It includes nurse-led MST and dietetic-led screening, whereby dietitians or their assistants re-screened patients whom nurses had scored as MST zero (not at-risk), but who dietitians suspected were at-risk.

The incremental cost per additional true case of malnutrition identified with e-MST self-screening compared to observed current practice (nurse-led MST screening and dietetic-led screening) and standard practice (nurse-led screening only) was estimated. This approach reflects the pragmatic nature of the study and the consideration that observed current practice within this study differed from the standard hospital procedure for nurse-led screening only. A case of malnutrition was estimated based on SGA diagnosis upon assessment from a dietitian after referral. The probability of being malnourished was estimated for patients referred to dietitians based on observed current referral practice for patients with an e-MST self-screening score of two or more, and for patients with a nurse-led MST score of two or more (based on those that were seen by a dietitian). Where the number of referrals from an MST approach based on a score of two or more (e-MST or nurse-led) was less than that of current practice, we modeled a scenario where the same number of referrals are made as current practice but with a preference for those with an MST of two or more and the subsequent referrals (i.e., the difference between observed practice and the MST ≥ 2 referrals) were made of those with an MST score of less than two. The cost of nurses administering the MST was based on estimates of nurse time and the cost of administering the e-MST was based on the proportion of patients who required assistance in completing the tool and the estimated time of assisting patients. This time was costed based on the direct and indirect salary costs of a nurse (NG5, Queensland Nursing Enterprise Agreement [33]). Time spent by dietitians on inappropriate referrals was also considered. For a sub-sample of patients seen by a dietitian (*n* = 17), dietitians were asked to consider if this was an appropriate referral. The probability of inappropriate referral to the dietitian was estimated for each MST approach and included in the cost based on the estimated time per inappropriate referral of a dietitian and the direct and indirect salary cost of a dietitian (HP4, Queensland Allied Health Enterprise Agreement [34]). Probabilistic sensitivity analyses used Monte Carlo methods to characterize the uncertainty with 10,000 iterations, drawing an estimate for each cost and effectiveness parameter drawn from each parameter’s distribution. Gamma distributions were used for cost parameters and beta distributions for probability parameters. For transparency in reporting the costs of all the inputs, they were itemized and tabulated alongside the final calculated incremental cost.

## 3. Results

### 3.1. Demographic and Clinical Nutrition Data

A total of 121 patients were recruited for the study. Just over half (*n* = 63, 52.1%) were male and the median age was 59 (IQR 48–69; range 19–89) years. Table 2 details the wards, comorbidities, and nutrition data relating to participants. Figure 1 depicts patient flow through the study.

### 3.2. Feasibility

Both feasibility criteria were met. In total, 121 of 156 patients (78%) who were approached for recruitment consented to the study, exceeding this criterion (i.e., ≥50% of patients approached provided consent). All patients (100%) completed the e-MST within 24 h of recruitment (most immediately after recruitment). Of these, 118 e-MST scores (98%) were extracted from the electronic food service system (three scores were lost due to a technical error whereby patient e-MST data were reset).

Other feasibility data indicated that most patients (*n* = 72; 59%) could complete the e-MST with less than a minute’s explanation from the RA. Eight patients (7%) completed the e-MST with no explanation. The remaining 41 patients (34%) required some assistance or explanation from the RA. On average, the RA spent 2.3 min assisting these patients. Five participants’ family members assisted or completed the e-MST on behalf of the patient.

Patient-reported e-MST scores did not correlate well with nurse-conducted e-MST scores (i.e., scores documented by nurses in routine practice), with poor agreement overall (Kappa 0.091, *p* = 0.076). Nurses tended to document lower scores, with only 23% of patients screened as at-risk, while patient-reported scores tended to be higher (36% screened themselves as at-risk). Patient-completed e-MSTs were highly predictive of malnutrition (diagnosed by dietitians via the SGA, available for 33 patients) with a sensitivity of 85.7%, compared to nurse-completed MSTs (sensitivity of 59.1%). Specificity was higher for nurse-completed MSTs (83.3%) than patient-completed e-MSTs (66.7%).

In the second phase of data collection, where flyers were placed on patient meal trays providing instructions on how to complete the e-MST, only 12.6% of patients across the four study wards completed the e-MST. Generally, patients showed little interest in the flyers and despite a brief prompt from the RA, few patients completed the e-MST in this phase which provided minimal support or prompting.

### 3.3. Acceptability

Most patients said they had a positive (*n* = 64, 52.9%) or very positive (*n* = 37, 30.6%) experience completing the e-MST (*n* = 20, 16.5% neutral). The majority found it very easy (*n* = 59, 48.8%) or easy (*n* = 52, 43.0%) to follow the instructions on their bedside computer screen (*n* = 5 neutral, *n* = 4 difficult, *n* = 1 very difficult). Patients were very satisfied (*n* = 76, 62.8%) or satisfied (*n* = 42, 34.7%) with the explanations/assistance provided by research personnel in completing the nutrition screen (*n* = 2 neutral). Most patients said it was no burden at all (*n* = 81, 67.5%) or not much burden (i = 25, 20.8%) to complete the e-MST on their bedside computer (i = 10 neutral, *n* = 3 somewhat burdensome, *n* = 1 very burdensome).

Acceptability interviews were conducted with seven patients and one family member. Data were organized into two main themes with several subthemes, described in Table 3.

### 3.4. Cost Effectiveness

The cost analysis (Table 4) shows the total costs per patient screened and the total cost per true case of malnutrition identified. Observed current practice identified 22 malnourished cases from 33 referrals, whereas standard practice (nurse MST ≥ 2) and patient-led MST ≥ 2 were estimated to yield 27 and 28.5 cases of malnutrition, respectively. With all cost inputs considered, the observed current practice cost was AUD $17.99 per patient compared to the standard practice (AUD $16.06) and patient-led MST ≥ 2 approach (AUD $14.76). The patient-led approach was estimated to yield the most malnourished cases (28.5 cases) and no inappropriate referrals. This was greater than both observed current practice (22 malnourished cases and 5.8 inappropriate referrals) and standard practice (27 malnourished cases and 5.5 inappropriate referrals). The incremental cost per additional malnourished case for patient-led e-MST compared to observed current practice was a saving of $0.50 per patient, which can be interpreted as patient-led screening yielding more true malnutrition cases and costing less than observed current practice to do so. Interestingly, nurse screening for 33 modeled referrals (based on standard practice) was also predicted to yield more true malnutrition cases and was less costly than observed current practice, however, more expensive than patient-led screening (Table 4). Appendix A present the probabilistic sensitivity analysis results for the incremental cost per patient seen using patient-led screening (3.23 AUD saving/patient) and nurse-led screening (1.93 AUD saving/patient), respectively.

## 4. Discussion

This study evaluated the feasibility, acceptability, and clinical and potential cost-effectiveness of patient-led, technology-assisted nutrition screening using the MST. Overall, this approach was found to be feasible, acceptable to patients and cost-effective for the health service, yielding more true cases of malnutrition and being less costly compared to current practice.

Patient-led eMST screening was shown to be highly feasible. The majority (78%) of patients approached for this study agreed to take part, and all recruited patients completed the e-MST, suggesting that a significant proportion of patients may agree to technology-assisted self-screening for malnutrition risk in practice. This recruitment rate is consistent with a study evaluating patient completion of a paper or app version of the Patient-Generated Subjective Global Assessment Short Form among inpatients with head and neck cancer (75% recruitment rate) [35] and higher than a study of electronic self-screening among hospital outpatients using the Malnutrition Universal Screening Tool (63% recruitment) [23]. It should be noted that only 70% of patients on study wards were considered eligible for this study (however, some were excluded due to prior participation in the study), so it could be estimated that around half of the patients on these study wards would be considered able (i.e., eligible) and willing (i.e., consenting) to complete their own e-MST. If half of the patients on a ward could self-screen using bedside technology with minimal assistance, this would be of huge benefit to nurses and dietitians and would exceed some of the screening rates observed in practice and reported in the literature. For example, a 2020 study of 38 Portuguese hospitals reported that only 28% of inpatients were screened for malnutrition within 48 h of admission [36]. We acknowledge that electronic self-screening may not be appropriate for all patients at all times and in different settings, for example, those from non-English speaking backgrounds or those with cognitive or physical impairments (e.g., dementia, low vision, poor motor skills), as one of our interview participants also noted. In some cases, there may be capacity for family members to assist—in our study, five participants’ family members completed the e-MST for them. There is also the potential for dietetic assistants to undertake malnutrition risk screening, with growing evidence suggesting that role delegation to dietetic assistants may improve nutrition care delivery, as well as patient and workforce outcomes [37]. This could result in further cost savings if paired with the patient-led approach tested in this study (i.e., dietetic assistants helping patients who need it) and protect higher clinical value front-line nursing time.

All recruited patients completed the e-MST and most found it easy to do. The majority (59%) completed the e-MST with less than a minute’s explanation, and those who asked for support needed an average of two minutes of assistance. Considering that nurses reported spending between 1–2 min on patient MSTs, it seems feasible to adopt the e-MST within usual practice, with nurses supporting patients who need it. Previous research has found that patients are agreeable to participating in nutrition care activities via technology during hospitalization, but value human interaction to support/assist them in using the technology [24], which is consistent with the findings from our patient interviews. Studies from Australian hospitals also report that hospital staff, including dietitians, nurses and executives, value the use of technology in supporting patient care and creating efficiency gains in practice [19,38]. In fact, staff at the current study’s hospital site who were previously interviewed about the patient-led e-MST during its development were complimentary of this technology, with nurses reporting it could save them time and paperwork and dietitians believing it could improve MST accuracy [19]. Staff also liked the fact that it engaged patients in their nutrition care [19].

An important finding from this study was the poor accuracy observed in nurse-completed MST scores. Patients tended to score themselves as at-risk more often than nurses did, indicating a practice issue that requires further exploration. Anecdotally, this can be described in the local setting to be due to the introduction of an electronic medical records system at the study hospital several years ago, with all screening, assessment, observations, etc., now recorded in electronic forms. An unintended consequence of this has been that many nurses now complete the MST at their workstation rather than the patient’s bedside, so patients are not always asked the MST questions directly. Instead, nurses may select scores on an ad-hoc basis or by clinical judgment. Most often, nurses tick either ‘no’ to both MST questions, resulting in a score of ‘0’ (not at risk), or will tick ‘unsure’ for the weight loss question, resulting in a score of ‘2’ (at risk), which triggers an automatic dietitian referral. Anecdotally, dietitians report that many at-risk patients are not detected (with a false negative score of 0) and many patients not at-risk are automatically referred (with a false positive score of 2), resulting in dietitians spending ~100 min per week on inappropriate MST 2 referrals, and dietetic assistants spending additional time screening for malnutrition risk in patients who have MST 0 documented by nurses but who dietitians suspect are at-risk. This issue may also explain the discrepancy between the sensitivity and specificity findings, where sensitivity was higher for patient-led e-MST (better detection of true positives, as patients were more likely to screen themselves as at-risk) but specificity was higher in nurse-led MST (higher detection of true negatives, as nurses were more likely to score patients as not at-risk). Clinically, the lower specificity is less of a concern in a screening tool and the higher sensitivity is preferred to ensure at-risk patients are detected.

Patient-led screening may be more cost-effective compared to observed current practice (which includes nurse-led and dietetics-led screening) and standard practice (nurse-led screening). The results of our cost-effectiveness study are limited by the modeled data required for the nurse-led screening (which is considered standard practice according to the hospital’s screening policy). However, patient-led screening yielded more true malnutrition cases identified for the 33 case referrals received in current practice and was less costly with all the cost inputs considered. The main driver of the cost differential was nurse time saved and no cost of inappropriate referrals, where dietitians would spend time reviewing referrals and triaging importance. While this difference may seem small, this is based on a small patient sample (*n* = 121 patients) and 33 referrals. Given the study hospital has around 140,000 admissions per year (2021–2022) [39], there is substantial potential for patient-led MST screening to be significantly cost-saving compared to observed current practice and standard practice over time, and when considering the critical gap between the number of patients typically screened compared vs. recommended to be screened [15]. It was beyond the scope of this study to estimate the health and monetary benefits associated with the increased yield in true malnutrition cases being seen by a dietitian. However, it is important to note that funding is typically received by health services depending on case mix complexity (along with a number of other factors), with malnutrition being one of these. Therefore, health services would be eligible for the increased reimbursement associated with greater case detection. While there are no recent estimates of reimbursements received for malnutrition by Australian hospitals, a 2015 study [40] using 2005 cost data [41] estimated that documented malnutrition changed case-mix funding for 20% of patients, with around AUD $3500 of hypothetical reimbursement estimated for each patient. In addition to case mix funding for hospitals, increasing the number of malnourished patients seen by dietitians provides the opportunity to implement effective nutrition interventions for these patients, to potentially reduce the cost of their hospital attendance and improve health outcomes. Inpatient nutrition care is shown to be highly cost-effective to reduce risks for intensive care admissions and hospital-associated complications, while also improving patient survival [8]. In England, it was estimated that the annual costs of screening, assessment, and nutritional support in hospitals are more than offset by decreased health care use and costs, with a net saving of ~£65 million annually [12]. If these were to be considered, it is likely that patient-led MST screening would be even more cost-beneficial to both the health service and the health system more broadly, as it yields more true malnutrition cases compared to current observed and standard practice.

### 4.1. Strengths and Limitations

This study has some key strengths worth noting. It was conducted in a real-world hospital setting on wards where patients present with, and are discharged at risk of, malnutrition. This provides great insight into how patient-led MST screening would work in practice. The study includes outcomes from both the health service (i.e., feasibility, clinical characteristics and costs) and patient perspective (i.e., acceptability), which can be used by health services, researchers and clinicians when considering how technology can support effective and efficient malnutrition screening using the MST.

There are important limitations to note. Firstly, this is a feasibility study without an active or historical control arm. This means that all analyses comparing patient-led screening to nurse screening need to be confirmed in an adequately powered effectiveness study with a control arm. Second, we only recruited patients able to communicate in English and cognitively intact, so the completion rates and cost inputs (including clinician time to explain and assist with using the technology) reflect this population studied. Third, participants completing this study were under research-controlled conditions, including consent and study explanation (i.e., how to complete the screening), so they had been exposed to a lot of prompting to complete the e-MST. As patients could request assistance from a research assistant (and this information was collected and included in costings), it is important to note that this might not always be available in usual practice. Fourth, the malnourished cases yielded from the referrals and associated with MST scores may not be representative of the yield among those with the same MST score but not seen by the dietitian. For the cost-effectiveness study, it was assumed that the same yield of malnourished cases would be observed among those that were not seen by a dietitian but had the same MST score. Where this does not hold true, the number of malnourished cases will differ. A larger powered study with a control group may assist in gathering and accounting for these limitations in the future. There may be a benefit of future research considering implementation designs with the ability to adapt to changing processes and engagement (e.g., stepped-wedge implementation trial and/or adaptive trial designs with optimization strategies).

### 4.2. Implications for Practice

Involving patients in their nutrition care has many benefits such as improved engagement with nutrition interventions, ensuring care plans are tailored to the individual, and an appreciation for how the care plan may change from the hospital to home. However, there are implementation considerations to consider before the patient-led e-MST screening can be trialed as part of routine clinical practice in the health service. Implementation requires some visual and usability adjustments to the e-MST. There is also more work to be conducted to effectively engage end-users (nurses, dietitians) who are the critical adopters of the technology and workflow. If they do not see the value proposition or benefit to patients, then its success and sustainable uptake are unlikely. Training and familiarisation for clinicians and patients is required, including how to use the technology and how to prompt patients to complete it. As our participants highlighted, they valued the RA prompting and assisting them, so there will always be an element of human interaction required to implement and sustain a technology-supported MST screening workflow. This is also confounded by individual digital literacy and familiarization, affecting both patients and clinicians. These are elements that previous research has shown to be critical from patient perspectives for improving digital engagement and overall experience of care [42].

## 5. Conclusions

In conclusion, patient-led e-MST screening was found to be feasible, acceptable to patients, and cost-effective (more malnutrition yield and less costly) compared to current practice for the health service. This approach was highly feasible, and all consenting patients completed the screening. Only 70% of patients were eligible to participate, so standard practice screening still has an important role to play for patients who do not speak English, are not overly digitally literate, or would prefer not to complete their screening of malnutrition risk themselves. This study can be used to inform the suitability of technology-supported nutrition screening. Future research should compare the clinical and cost-effectiveness of standard practice MST screening to patient-led, technology-supported MST screening.

## Figures and Tables

**Figure 1 nutrients-16-01139-f001:**
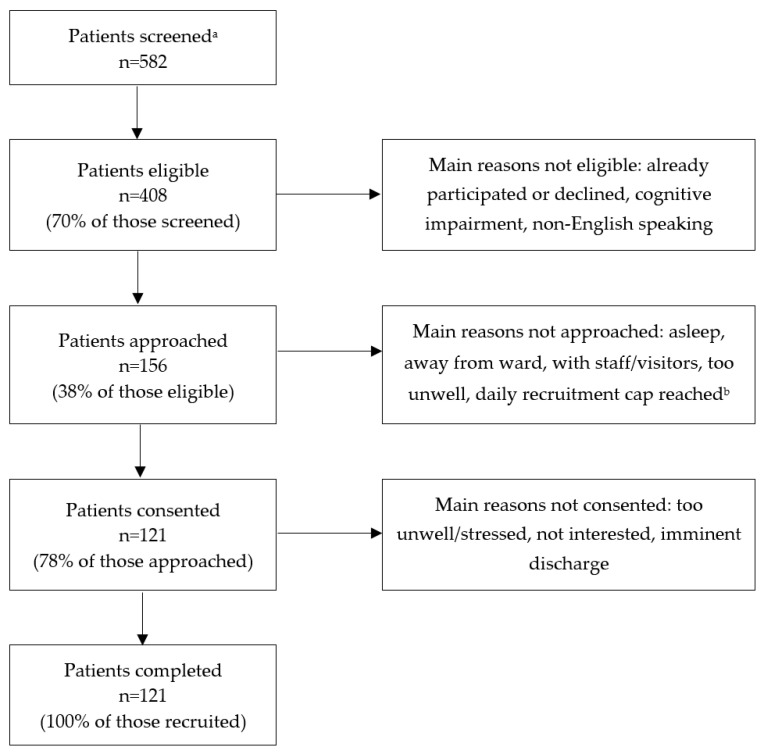
Patient flow diagram. ^a^ Includes all patients on study wards on recruitment days. ^b^ One research assistant was employed for the study and could recruit up to ten patients per day.

**Table 1 nutrients-16-01139-t001:** Data collection schedule.

Study Phase	Study Procedure	Data Collection
Main study (Oct–Nov 2022)	Eligibility screening and recruitment (participants shown e-MST on bedside computer by RA)	Recruitment rate
	Participant completes e-MST within 24 h of recruitment (with/without assistance from RA)	e-MST completion date/time and score (extracted from EFS), RA time spent assisting patients
	RA administers survey (same day as patient completes e-MST) and interviews a subset of participants	Patient satisfaction survey and acceptability semi-structured interviews
	Electronic medical records data extraction	Demographic, clinical and nutrition data (including nurse-completed MST scores)
	Economic evaluation of e-MST screening per true case of malnutrition	Surveys (nurses, dietitians, dietitian assistants, RA), ieMR and EFS MST scores, Queensland Health standard award rates (2023 for dietitian (HP4), nurse (NG5), dietitian assistant (CA3), RA (HP 3.4)
Sustainability study (April 2023)	Flyers sent on patient meal trays (five wards; one ward per day)	RA field notes
	Patients complete e-MST by end of day	e-MST completion and scores extracted from EFS

EFS: Electronic foodservice system; e-MST: Electronic Malnutrition Screening Tool; ieMR: integrated electronic medical record, MST: Malnutrition Screening Tool; RA: Research assistant.

**Table 2 nutrients-16-01139-t002:** Participant demographics and clinical nutrition data.

Variable		N (%)
Ward	Oncology Orthopedic Gastrointestinal surgical Gastrointestinal medical Surgical Renal Medical/vascular	24 (19.8) 21 (17.4) 21 (17.4) 17 (14.0) 17 (14.0) 12 (9.9) 9 (7.4)
Comorbidities	Cardiovascular Gastrointestinal Cancer Surgery Respiratory Diabetes Renal Neurology Trauma	53 (43.8) 46 (38.0) 42 (34.7) 39 (32.2) 21 (17.4) 21 (17.4) 20 (16.5) 10 (8.3) 7 (5.8)
Dietetic input	Seen by dietitian during admission Seen by dietetic assistant	55 (45.4) 33 (27.3)
Nutrition support	Oral nutrition supplements Enteral/parenteral nutrition	24 (19.8) 3 (2.5)
Prescribed diet	High protein Regular Texture modified Other	64 (52.9) 15 (12.4) 10 (8.3) 32 (26.4)
Nutrition status ^1^	Well nourished Mildly to moderately malnourished Severely malnourished	11 (33.3) 17 (51.5) 5 (15.1)

^1^ As per Subjective Global Assessment completed by ward dietitian; available for 33 patients (extracted from electronic medical record).

**Table 3 nutrients-16-01139-t003:** Thematic analysis of patient perceptions of the e-MST.

Theme	Sub-Theme	Description and Quotes
Using technology for nutrition screening	Easy to complete but prompts may be required	Patients found the e-MST straightforward and “*easy to do*” (P120). Most felt confident completing it and would do so again in future admissions. Patients said the e-MST posed a minimal burden and did not impact their day, as “you’ve got to do your dinners anyway” (P94). Some indicated a prompt may be needed for patients to complete it. - *I think it’s a lot easier on there (screen) because it’s very quick and if a nurse comes and points out, can you just push this button and answer those two questions—it’s done, easy.* P119 - *It’s easy enough to see because you told me it was there, but if I didn’t know it was there, I wouldn’t press it… let it be flashing on the screen… ‘please complete this’.* P121 *There was a button there… I clicked on that… just out of curiosity… no prompting or whatever was needed… I just did it for fun. And there’s a lot of people that probably do it for fun, potentially. It gets boring in hospital.* P34
	Enhancing technology usability for patients	Some patients said they faced no challenges in using the bedside computer, while others described issues they themselves faced, or perceived other patients may face when using the technology. Some thought age, unfamiliarity with technology, being acutely unwell, or conditions like poor motor control and low vision could be barriers to other patients completing the e-MST. - *Bigger buttons… possibly colour coordinated buttons… high contrast for people with poor vision*. P34 *Older folks especially can be quite technologically unsavvy, and so they might struggle.* P34
	Perceived benefits of electronic nutrition screening in healthcare	Several patients thought patient-led screening could benefit hospital staff by saving them time and providing information for care purposes. Some also thought it could help prompt care-related conversations that benefit patients. - *I think the patient should do it [e-MST]… just one less thing that they [nurses] have to worry about, and they can use that in your treatment.* P98 *I think it is a good thing… nurses are great, but they don’t actually know what’s going on inside someone’s brain, so they don’t know if they have a decreased appetite… it is good, I think for patients to be given the chance to complete it.* P34
Patient perceptions of nutrition screening in hospital	Patient understanding and value of the MST	Patients had mixed understandings of the MST as a screening tool and thought providing more information would be helpful, including its purpose and scoring, interpreting the MST questions, and the impact of the survey results on their healthcare. Some patients understood the purpose of the MST while others wanted more clarity: - *I do think it would be helpful for the individuals who are doing it to know that there is a score… because someone might not know that they have a risk of malnutrition.* P34 *It would probably be good to know… something like “we will review your answers and the dietitian may see you”, just so that people can be forewarned… someone might be coming and seeing me.* P34
	Personal context is important in nutrition risk screening	Patients spoke about the importance of their personal context and how this affected their interest in completing the e-MST. For example, one patient thought nutrition was “*an issue in everyone’s life*” (P121) and another “*understood the context of [nutrition screening]*” (P119) from working in aged care. Both these patients expressed appreciation for the screening tool and its value to health professionals. However, some patients thought the MST didn’t account for certain personal contexts such as the ability to access food and monitor their weight: - *I was homeless before coming into hospital and so one of the questions asks…if you lost weight, and I don’t know what my weight was before coming into hospital.* P34 *I’m eating fine because I haven’t changed what I’m eating. However, I can get food… there’s a lot of people that can’t afford meals that I see [in aged care].* P119

**Table 4 nutrients-16-01139-t004:** Cost-effectiveness analysis showing the cost per true case of malnutrition identified. All costs are presented in Australian Dollars (AUD).

Model Input	Observed Current Practice ^1^ (N = 121)	Nurse-Led MST ≥2 (N = 121)	Patient-Led e-MST ≥2 (N = 121)
Referral pathway			
Patients requiring nurse input, *n* (%)	121 (100)	121 (100)	41 (33.9)
Nurse time per occasion of input, minutes	1.2	1.2	2.3
Nurse cost per minute, AUD	$0.93	$0.93	$0.93
MST nurse cost, AUD	$134.24	$134.24	$87.38
Patients identified from additional dietetic-led screening ^2^, *n* (%)	16 (13)	5 (4)	0 (0)
Dietitian time per patient identified, minutes	10	10	0
Dietitian cost per minute, AUD	$1.42	$1.42	$1.42
Cost for dietitian-identified patient, AUD	$227.20	$50.00	$ -
Total cost of referral pathway, AUD	$361.44	$134.24	$87.38
Cost per patient of referral pathway, AUD	$2.99	$1.11	$0.72
Dietitian attendances			
Referrals, n	33	33 ^3^	33
Cost of dietitian attendance, AUD	$1698.66	$1698.66	$1698.66
Proportion of referrals inappropriate, %	17.6%	16.7%	0.0%
Estimated referrals inappropriate, *n*	5.8	5.5	-
Dietitian time wasted per inappropriate referral (false positive MSTs), minutes	14.17	14.17	14.17
Cost of inappropriate referrals, AUD	$117.16	$110.65	$ -
Total cost, AUD	$2177.26	$1943.55	$1734.59
Cost per patient, AUD	$17.99	$16.06	$14.76
Incremental cost per patient, AUD	REF	−$1.93	−$3.23
Malnourished patients per referral, %	66.7%	81.8%	86.4%
Estimated total number of malnourished patients, n	22.0	27.0	28.5
Incremental number of malnourished patients, *n*	REF	5.0	6.5
Incremental cost per additional malnourished patient attended, AUD	REF	−$0.39	−$0.50

AUD: Australian Dollars, MST: Malnutrition screening tool, REF: Reference. ^1^ Dietitians estimated spending ~100 min per week per ward on inappropriate (false positive) MST referrals. Nurses reported spending 0.5–2 min per patient on MST in current observed practice. ^2^ Standard practice should involve nurse-led MST screening and referral only, but observed current practice involved dietitians spending additional time re-screening patients with false-negative nurse-led MST scores. ^3^ Comprised of 28 patients with a nurse-led MST score of ≥2 and five patients with a nurse-led MST score of 0–1; predicted malnourished cases were estimated based on MST score for those seen by a dietitian.

## Data Availability

The data presented in this study are available on request from the corresponding author due to ethical restrictions on data sharing.

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
