# Peer review of "Patient-Led, Technology-Assisted Malnutrition Risk Screening in Hospital: A Feasibility Study"

_nutrients, 2024, doi:10.3390/nu16081139_

Round 1

Reviewer 1 Report

Comments and Suggestions for Authors

General comments:

This is a very interesting study that is done well.

Minor Concerns

Table 1 needs some lines or spacing added to provide clarity about which study procedures and data collection items belong in which phase of the study. For example, is Economic evaluation of e-MST screening per true case of malnutrition part of the Main study phase or part of the Sustainability study phase? Also, were RA filed notes collected during the Main study phase or the Sustainability study phase? Maybe a timeline would be more appropriate than this presentation of information.

Table 2 needs some lines or spacing added to provide clarity about which items listed in the second and third columns belong to which variable listed in the first column.

Table 3 needs some lines or spacing added to provide clarity about which items listed in the second and third columns belong to which Theme listed in the first column.

In the section entitled “3.4 Cost Effectiveness” on page 10 Lines 290-308, it would be helpful to use the nomenclature defined in the section entitled “2.4. Data Analysis” on page 5 Lines 194-209. Specifically, the term Standard Practice appears to be the same as nurse-led MST screening. Otherwise, it is a bit confusing to follow which is which in the analysis.

Major Concerns

None

Author Response

The authors would like to thank Reviewer 1 for their time and expertise reviewing this paper. We have provided our response to your queries in the attached document, and have updated the manuscript accordingly.

Reviewer 2 Report

Comments and Suggestions for Authors

Thank you for the opportunity to review the feasibility study aims to evaluate the acceptability and potential cost- effectiveness of patient-led, technology-assisted nutrition screening in hospital. While it is an interesting piece, there are several considerations that need to be addressed. 

1. The items and evaluation methods of The Malnutrition Screening Tool were not mentioned. Please clarify MST, MSTs and e-MST and give the reliability and validity. Are there any differences among the content of tool/surveying time/precautions?

2. what is ”usual care, please give more in detail.

3. The time for calculating the economic costs in nurses should be the nurse-led evaluating time minuses the estimated time of assisting patients. Why did included the proportion of patients who required assistance in completing the tool?

4. Please give the questions for Patient interviews methodology.

Results

5. Please check it. You may delete the pane on the right.

 6. As for the purpose of economic costs, Table 4 is too specific. If for the other purpose, please give more clarification in methodology.

7. In this study, Patient-completed e-MSTs were highly predictive of malnutrition with a sensitivity of 85.7% (sensitivity 59.1% in nurse-led one); while Specificity was higher for nurse completed MSTs (83.3%) than patient-completed e-MSTs (66.7%). How do you solve this  contradiction and recommend in clinical practice? 

Author Response

The authors would like to thank Reviewer 2 for their time and expertise reviewing this paper. We have provided our response to your queries in the attached document, and have updated the manuscript accordingly.
